# Scaling-up Importance Sampling for Markov Logic Networks

**Deepak Venugopal**
Department of Computer Science
University of Texas at Dallas
`dxv021000@utdallas.edu`

**Vibhav Gogate**
Department of Computer Science
University of Texas at Dallas
`vgogate@hlt.utdallas.edu`

## Abstract

Markov Logic Networks (MLNs) are weighted first-order logic templates for generating large (ground) Markov networks. Lifted inference algorithms for them bring the power of logical inference to probabilistic inference. These algorithms operate as much as possible at the compact first-order level, grounding or propositionalizing the MLN only as necessary. As a result, lifted inference algorithms can be much more scalable than propositional algorithms that operate directly on the much larger ground network. Unfortunately, existing lifted inference algorithms suffer from two interrelated problems, which severely affects their scalability in practice. First, for most real-world MLNs having complex structure, they are unable to exploit symmetries and end up grounding most atoms (the grounding problem). Second, they suffer from the *evidence problem*, which arises because evidence breaks symmetries, severely diminishing the power of lifted inference. In this paper, we address both problems by presenting a scalable, lifted importance sampling-based approach that never grounds the full MLN. Specifically, we show how to scale up the two main steps in importance sampling: sampling from the proposal distribution and weight computation. Scalable sampling is achieved by using an informed, easy-to-sample proposal distribution derived from a compressed MLN-representation. Fast weight computation is achieved by only visiting a small subset of the sampled groundings of each formula instead of all of its possible groundings. We show that our new algorithm yields an asymptotically unbiased estimate. Our experiments on several MLNs clearly demonstrate the promise of our approach.

## 1 Introduction

Markov Logic Networks (MLNs) [5] are powerful template models that define Markov networks by instantiating first-order formulas with objects from its domain. Designing scalable inference for MLNs is a challenging task because as the domain-size increases, the Markov network underlying the MLN can become extremely large. Lifted inference algorithms [1, 2, 3, 7, 8, 13, 15, 18] try to tackle this challenge by exploiting symmetries in the relational representation. However, current lifted inference approaches face two interrelated problems. First, most of these techniques have the *grounding problem*, i.e., unless the MLN has a specific symmetric, *liftable* structure [3, 4, 9], most algorithms tend to ground most formulas in the MLN and this is infeasible for large domains. Second, lifted inference algorithms have an *evidence problem*, i.e., even if the MLN is liftable, in the presence of arbitrary evidence, symmetries are broken and once again, lifted inference is just as scalable as propositional inference [16]. Both these problems are severe because, often, practical applications require arbitrarily structured MLNs which can handle arbitrary evidence. To handle this problem, a promising approach is to approximate/bias the MLN distribution such that inference is less expensive on this biased MLN. This idea has been explored in recent work such as [16] which uses the idea of introducing new symmetries or [19] which uses unsupervised learning to reduce the objects in the

domain. However, in both these approaches, it may turn out that for certain cases, the bias skews the MLN distribution to a large extent. Here, we propose a general-purpose importance sampling based algorithm that retains the scalability of the aforementioned biased approaches but has theoretical guarantees, i.e., it yields asymptotically unbiased estimates.

Importance sampling, a widely used sampling approach has two steps, namely, we first sample from a proposal distribution and next, for each sample, we compute its *importance weight*. It turns out that for MLNs, both steps can be computationally expensive. Therefore, we scale-up each of these steps. Specifically, to scale-up step one, based on the recently proposed MLN approximation approach [19], we design an informed proposal distribution using a "compressed" representation of the ground MLN. We then compile a symbolic counting formula where each symbol is *lifted*, i.e., it represents multiple assignments to multiple ground atoms. The compilation allows us to sample each lifted symbol efficiently using Gibbs sampling. Importantly, the state space of the sampler depends upon the number of symbols allowing us to trade-off accuracy-of-the-proposal with efficiency.

Step two requires iterating over all ground formulas to compute the number of groundings satisfied by a sample. Though this operation can be made space-efficient (for bounded formula-length), i.e., we can go over each grounding independently, the time-complexity is prohibitively large and is equivalent to the grounding problem. For example, consider a simple relationship, $\texttt{Friends}(x, y) \wedge \texttt{Likes}(y, z) \Rightarrow \texttt{Likes}(x, z)$. If the domain-size of each variable is 100, then to obtain the importance weight of a single sample, we need to process 1 million ground formulas which is practically infeasible. Therefore, to make this weight-computation step feasible, we propose the following approach. We use a second sampler to sample ground formulas in the MLN and compute the importance weight based on the sampled groundings. We show that this method yields asymptotically unbiased estimates. Further, by taking advantage of first-order structure, we reduce the variance of estimates in many cases through *Rao-Blackwellization* [11].

We perform experiments on varied MLN structures (Alchemy benchmarks [10]) with arbitrary evidence to illustrate the generality of our approach. We show that using our approach, we can systematically trade-off accuracy with efficiency and can scale-up inference to extremely large domain-sizes which cannot be handled by state-of-the-art MLN systems such as Alchemy.

## 2 Preliminaries

### 2.1 Markov Logic

In this paper, we assume a strict subset of first-order logic called *finite Herbrand logic*. Thus, we assume that we have no function constants and finitely many object constants. We also assume that each argument of each predicate is typed and can only be assigned to a fixed subset of constants. By extension, each logical variable in each formula is also typed. The domain of a term $x$ in any formula refers to the set of constants that can be substituted for $x$ and is represented as $\Delta_x$. We further assume that all first-order formulas are disjunctive (clauses), have no free logical variables (namely, each logical variable is quantified), have only universally quantified logical variables (CNF). Note that all first-order formulas can be easily converted to this form. A ground atom is an atom that contains no logical variables.

Markov logic extends FOL by softening the hard constraints expressed by the formulas. A soft formula or a weighted formula is a pair $(f, w)$ where $f$ is a formula in FOL and $w$ is a real-number. A MLN denoted by $\mathcal{M}$, is a set of weighted formulas $(f_i, w_i)$. Given a set of constants that represent objects in the domain, an MLN defines a Markov network or a log-linear model. The Markov network is obtained by grounding the weighted first-order knowledge base and represents the following probability distribution.

$$P_{\mathcal{M}}(\omega) = \frac{1}{Z(\mathcal{M})} \exp\left(\sum_i w_i N(f_i, \omega)\right) \qquad (1)$$

where $\omega$ is a world, $N(f_i, \omega)$ is the number of groundings of $f_i$ that evaluate to $\texttt{True}$ in the world $\omega$ and $Z(\mathcal{M})$ is a normalization constant or the partition function.

In this paper, we assume that the input MLN to our algorithm is in normal form [9, 12]. A *normal* MLN [9] is an MLN that satisfies the following two properties: (1) There are no constants in any formula, and (2) If two distinct atoms with the same predicate symbol have variables $x$ and $y$ in

the same position then $\Delta_x = \Delta_y$. An important distinction here is that, unlike in previous work on lifted inference that use normal forms [7, 9] which require the MLN along with the associated evidence to be normalized, here we only require the MLN in normal form. This is important because normalizing the MLN along with evidence typically requires grounding the MLN and blows-up its size. In contrast, normalizing without evidence typically does not change the MLN. For instance, in all the benchmarks in Alchemy, the MLNs are already normalized.

Two main inference problems in MLNs are computing the partition function and the marginal probabilities of query atoms given evidence. In this paper, we focus on the latter.

## 2.2 Importance Sampling

Importance sampling [6] is a standard sampling-based approach, where we draw samples from a proposal distribution $H$ that is easier to sample compared to sampling from the true distribution $P$. Each sample is then weighted with its *importance weight* to correct for the fact that it is drawn from the wrong distribution. To compute the marginal probabilities from the weighted samples, we use the following estimator.

$$P'(\bar{Q}) = \frac{\sum_{t=1}^{T} \delta_{\bar{Q}}(\bar{\mathbf{s}}^{(t)}) w(\bar{\mathbf{s}}^{(t)})}{\sum_{t=1}^{T} w(\bar{\mathbf{s}}^{(t)})} \tag{2}$$

where $\bar{\mathbf{s}}^{(t)}$ is the $t^{th}$ sample drawn from $H$, $\delta_{\bar{Q}}(\bar{\mathbf{s}}^{(t)}) = 1$ iff the query atom $Q$ is assigned $\bar{Q}$ in $\bar{\mathbf{s}}^{(t)}$ and 0 otherwise, $w(\bar{\mathbf{s}}^{(t)})$ is the importance weight of the sample given by $\frac{P(\bar{\mathbf{s}}^{(t)})}{H(\bar{\mathbf{s}}^{(t)})}$.

$P'(\bar{Q})$ computed from Eq. (2) is an asymptotically unbiased estimate of $P_{\mathcal{M}}(\bar{Q})$, namely as $T \to \infty$ $P'(\bar{Q})$ almost surely converges to $P(\bar{Q})$. Eq. (2) is called as a ratio estimate or a normalized estimate because *we only need to know each sample's importance weight up to a normalizing constant*. We will leverage this property throughout the paper.

## 2.3 Compressed MLN Representation

Recently, we [19] proposed an approach to generate a "compressed" approximation of the MLN using unsupervised learning. Specifically, for each unique domain in the MLN, the objects in that domain are clustered into groups based on approximate symmetries. To learn the clusters effectively, we use standard clustering algorithms and a distance function based on the evidence structure presented to the MLN. The distance function is constructed to ensure that objects that are approximately symmetrical to each other (from an inference perspective) are placed in a common cluster.

Formally, given a MLN $\mathcal{M}$, let $\mathbb{D}$ denote the set of all domains in $\mathcal{M}$. That is, $\mathbf{D} \in \mathbb{D}$ is a set of objects that belong to the same domain. To compress $\mathcal{M}$, we consider each $\mathbf{D} \in \mathbb{D}$ independently and learn a new domain $\mathbf{D}'$ where $|\mathbf{D}'| \leq \mathbf{D}$ and $g : \mathbf{D} \to \mathbf{D}'$ is a surjective mapping, i.e., $\forall \mu \in \mathbf{D}'$, $\exists C \in \mathbf{D}$ such that $g(C) = \mu$. In other words, each cluster of objects is replaced by its cluster center in the reduced domain.

In this paper, we utilize the above approach to build an informed proposal distribution for importance sampling.

## 3 Scalable Importance Sampling

In this section, we describe the two main steps in our new importance sampling algorithm: (a) constructing and sampling the proposal distribution, and (b) computing the sample weight. We carefully design each step, ensuring that we never ground the full MLN. As a result, the computational complexity of our method is much smaller than existing importance sampling approaches [8].

### 3.1 Constructing and Sampling the Proposal Distribution

We first compress the domains of the given MLN, say $\mathcal{M}$, based on the method in [19]. Let $\hat{\mathcal{M}}$ be the network obtained by grounding $\hat{\mathcal{M}}$ with its reduced domains (which corresponds to the cluster centers) and let $\mathcal{M}_G$ be the ground Markov network of $\mathcal{M}$ using the original domains. $\hat{\mathcal{M}}$ and $\mathcal{M}_G$

**Formulas:**

$R(x) \lor S(x, y), w$

**Domains:**

$\Delta_x = \{A_1, B_1, C_1, D_1\}$
$\Delta_y = \{A_2, B_2, C_2, D_2\}$

(a)

**Formulas:**

$R_1(\mu_1) \lor S(\mu_1, \mu_3), w; R_1(\mu_2) \lor S(\mu_2, \mu_3), w$
$R_1(\mu_1) \lor S(\mu_1, \mu_4), w; R_1(\mu_2) \lor S(\mu_2, \mu_4), w$

**Domains:**

$\zeta(\mu_1) = \{A_1, B_1\}; \zeta(\mu_2) = \{C_1, D_1\}$
$\zeta(\mu_3) = \{A_2, B_2\}; \text{ and } \zeta(\mu_4) = \{C_2, D_2\}$

(b)

Figure 1: (a) an example MLN $\mathcal{M}$ and (b) MLN $\hat{\mathcal{M}}$ obtained from $\mathcal{M}$ by grounding each logical variable in $\mathcal{M}$ by the cluster centers $\mu_1, \ldots, \mu_4$.

are related as follows. We can think of $\hat{\mathcal{M}}$ as an MLN, in which the logical variables are the cluster centers. If we set the domain of each logical variable corresponding to cluster center $\mu \in \mathbf{D}'$ to $\zeta(\mu)$ where $\zeta(\mu) = \{C \in \mathbf{D}|g(C) = \mu\}$, then the ground Markov network of $\hat{\mathcal{M}}$ is $\mathcal{M}_G$. Figure 1 shows an example MLN $\mathcal{M}$ and its corresponding compressed MLN $\hat{\mathcal{M}}$. Notice that the Markov network obtained by grounding $\mathcal{M}$ is the same as the one obtained by grounding $\hat{\mathcal{M}}$.

Next, we describe how to generate samples from $\hat{\mathcal{M}}$. Let $\hat{\mathcal{M}}$ contain $\hat{K}$ predicates, for which we assume some ordering. Let $\mathbf{E}$ and $\mathbf{U}$ represent the counts of true (evidence) and unknown ground atoms respectively. For instance, $E_i \in \mathbf{E}$ represents the number of true ground atoms corresponding to the $i$-th predicate in $\hat{\mathcal{M}}$. To keep the equations more readable, we assume that we only have positive evidence (i.e., an assertion that the ground atom is true). Note that it is straightforward to extend the equations to the general case in which we have both positive and negative evidences.

Without loss of generality, let the $j$-th formula in $\hat{\mathcal{M}}$, denoted by $f_j$, contain the atoms $p_1, \ldots p_k$ where $p_i$ is an instance of the $p_i$-th predicate and if $i \leq m$, it has a positive sign else it has a negative sign. The task is to now count the total number of satisfied groundings in $f_j$ symbolically without actually going over the ground formulas. Unfortunately, this task is in $\#\mathcal{P}$. Therefore, we make the following approximation. Let $N(p_1, \ldots p_k)$ denote the number of the satisfied groundings of $f_j$ based on the assignments to all groundings of predicates indexed by $p_1, \ldots p_k$. Then, we will approximate $N(p_1, \ldots p_k)$ using $\sum_{i=1}^k N(p_i)$, thereby independently counting the number of satisfied groundings for each predicate. Clearly, our approximation overestimates the number of satisfied formulas because it ignores the joint dependencies between atoms in $f$. To compensate for this, we scale-down each count by a scaling factor ($\gamma$) which is the ratio of the actual number of ground formulas in $f$ to the assumed number of ground formulas. Next, we define these counting equations formally.

Given the $j$-th formula $f_j$ and a set of indexes $\mathbf{k}$, where $k \in \mathbf{k}$ corresponds to the $k$-th atom in $f_j$, let $\#G_{f_j}(\mathbf{k})$ denote the number of ground formulas in $f_j$ if all the terms in all atoms specified by $\mathbf{k}$ are replaced by constants. For instance, in the example shown in Fig. 1, let $f$ be $R_1(\mu_1) \lor S_1(\mu_1, \mu_3)$, then, $\#G_f(\emptyset) = 4$, $\#G_f(\{1\}) = 2$ and $\#G_f(\{2\}) = 1$. We now count $f_j$'s satisfied groundings symbolically as follows.

$$\mathcal{S}_j' = \gamma \sum_{i=1}^m E_{p_i} \#G_{f_j}(\{i\}) \tag{3}$$

where $\gamma = \frac{\#G_{f_j}(\emptyset)}{m\#G_{f_j}(\emptyset)} = \frac{1}{m}$ and $\mathcal{S}_j'$ is rounded to the nearest integer.

$$\mathcal{S}_j = \gamma \left( \sum_{i=1}^m \hat{S}_{p_i} \#G_{f_j}(\{i\}) + \sum_{i=m+1}^k (U_{p_i} - \hat{S}_{p_i}) \#G_{f_j}(\{i\}) \right) \tag{4}$$

where $\gamma = \frac{max(\#G_{f_j}(\emptyset) - \mathcal{S}_j', 0)}{k\#G_{f_j}(\emptyset)}$, $\hat{S}_{p_i}$ is a *lifted symbol* representing the total number of true ground atoms (among the unknown atoms) of the $p_i$-th predicate and $\mathcal{S}_j$ is rounded to the nearest integer.

The symbolic (un-normalized) proposal probability is given by the following equation.

$$H(\hat{\mathbf{S}}, \mathbf{E}) = \exp \left( \sum_{j=1}^C w_j \mathcal{S}_j \right) \tag{5}$$

**Algorithm 1:** Compute-Marginals

---

**Input**: $\hat{\mathcal{M}}$, $\zeta$, Evidence $\mathbf{E}$, Query $\mathbf{Q}$, sampling threshold $\beta$, thinning parameter $p$, iterations $T$
**Output**: Marginal probabilities $\mathcal{P}$ for $\mathbf{Q}$
**begin**
    Construct the symbolic counting formula Eq. (5)
    // Outer Sampler
    **for** *t = 1 to T* **do**
        Sample $\hat{\mathbf{S}}^{(t)}$ using Gibbs sampling on Eq. (5)
        After burn-in, for every $p$-th sample, generate $\bar{\mathbf{s}}^{(t)}$ from $\hat{\mathbf{S}}^{(t)}$
        **for** *each formula $f_i$* **do**
            // Inner Sampler
            **for** *c = 1 to $\beta$* **do**
                // Rao-Blackwellization
                $f_i'$ = Partially ground formula created by sampling assignments to shared variables in $f_i$
                Compute the satisfied groundings in $f_i'$
        Compute the sample weight using Eq. (7)
    Update the marginal probability estimates using Eq. (2)

---

where $C$ is the number of formulas in $\hat{\mathcal{M}}$ and $w_j$ is the weight of the $j$-th formula.

Given the symbolic equation Eq. (5), we sample the set of lifted symbols, $\hat{\mathbf{S}}$, using randomized Gibbs sampling. For this, we initialize all symbols to a random value. We then choose a random symbol $\hat{S}_i$ and substitute it in Eq. (5) for each value between 0 to $(\hat{U}_i)$ yielding a conditional distribution on $\hat{S}_i$ given assignments to $\hat{\mathbf{S}}_{-i}$, where $\hat{\mathbf{S}}_{-i}$ refers to all symbols other than the $i^{th}$ one. We then sample from this conditional distribution by taking into account that there are $\binom{\hat{U}_i}{v}$ different assignments corresponding to the $v^{th}$ value in the distribution, which corresponds to setting exactly $v$ groundings of the $i^{th}$ predicate to True. After the Markov chain has *mixed*, to reduce the dependency between successive Gibbs samples, we *thin* the samples and only use every $p$-th sample for estimation.

Note that during the process of sampling from the proposal, we only had to compute $\hat{\mathcal{M}}$, namely ground the original MLN with the cluster-centers. Therefore, the representation is *lifted* because we do not ground $\hat{\mathcal{M}}$. This helps us scale up the sampling step to large domains-sizes (since we can control the number of clusters).

### 3.2 Computing the Importance Weight

In order to compute the marginal probabilities as in Eq. (2), given a sample, we need to compute (up to a normalization constant) the weight of that sample. It is easy to see that a sample from the proposal (assignments on all symbols) has multiple possible assignments in the original MLN. For instance, suppose in our running example in Fig. 1, the symbol corresponding to $\mathtt{R}(\mu_1)$ has a value equal to 1, this corresponds to 2 different assignments in $\mathcal{M}$, either $\mathtt{R}(A1)$ is true or $\mathtt{R}(B1)$ is true. Formally, a sample from the proposal has $\prod_{i=1}^{\hat{K}} \binom{\hat{U}_i}{\hat{S}_i}$ different assignments in the true distribution. We assume that all these assignments are equi-probable (have the same weight) in the proposal. Thus, to compute the (un-normalized) probability of a sample w.r.t $\mathcal{M}$, we first convert the assignments on a specific sample, $\hat{\mathbf{S}}^{(t)}$ into one of the equi-probable assignments $\bar{\mathbf{s}}$ by randomly choosing one of the assignments. Then, we compute the (un-normalized) probability $P(\bar{\mathbf{s}}, \mathbf{E})$. The importance weight (upto a multiplicative constant) for the $t$-th sample is given by the ratio,

$$w(\hat{\mathbf{S}}^{(t)}, \mathbf{E}) = \frac{P(\bar{\mathbf{s}}^{(t)}, \mathbf{E})}{H(\hat{\mathbf{S}}^{(t)}, \mathbf{E})} \tag{6}$$

Plugging-in the weight computed by Eq. (6) into Eq. (2) yields an asymptotically unbiased estimate of the query marginal probabilities [11]. However, in the case of MLNs, computing Eq. (6) turns out to be a hard problem. Specifically, to compute $\hat{P}(\bar{\mathbf{s}}^{(t)}, \mathbf{E})$, given a sample, we need to go over each ground formula in $\mathcal{M}$ and check if it is satisfied or not. The *combined-complexity* [17] (domain-size as well as formula-size are assumed to be variable) of this operation for each formula

is #P-complete (cf. [5]). However, the *data complexity* (fixed formula-size, variable domain-size) is polynomial. That is, for $k$ variables in a formula where the domain-size of each variable is $d$, the complexity is clearly $O(d^k)$ to go over every grounding. However, in the case of MLNs, notice that a polynomial data-complexity is equivalent to the complexity of the grounding-problem, which is precisely what we are trying to avoid and is therefore intractable for all practical purposes. To make this weight-computation step tractable, we use an additional sampler which samples a bounded number of groundings of a formula in $\mathcal{M}$ and approximates the importance weight based on these sampled groundings. Formally,

Let $\mathbb{U}_i$ be a proposal distribution defined on the groundings of the $i$-th formula. Here, we define this distribution as a product of $|\mathbf{V}_i|$ uniform distributions where $\mathbf{V}_i = V_{i1} \ldots V_{ik}$ is the set of distinct variables in the $i$-th formula. Formally, $\mathbb{U}_i = \prod_{j=1}^{|\mathbf{V}_i|} \mathbb{U}_{ij}$, where $\mathbb{U}_{ij}$ is a uniform distribution over the domain-size of $V_{ik}$. A sample from $\mathbb{U}_i$ contains a grounding for every variable in the $i$-th formula. Using this, we can approximate the importance weight using the following equation.

$$w(\bar{\mathbf{s}}^{(t)}, \mathbf{E}, \bar{\mathbf{u}}_i^{(t)}) = \frac{\exp \left( \sum_{i=1}^M w_i \frac{N_i'(\bar{\mathbf{s}}^{(t)}, \mathbf{E}, \bar{\mathbf{u}}_i^{(t)})}{\beta \prod_{j=1}^{|\mathbf{V}_i|} \mathbb{U}_{ij}} \right)}{H(\hat{\mathbf{S}}^{(t)}, \mathbf{E})} \tag{7}$$

where $M$ is the number of formulas in $\mathcal{M}$, $\bar{\mathbf{u}}_i^{(t)}$ are $\beta$ groundings of the $i$-th formula drawn from $\mathbb{U}_i$ and $N_i'(\bar{\mathbf{s}}^{(t)}, \mathbf{E}, \bar{\mathbf{u}}_i^{(t)})$ is the count of satisfied groundings in $\bar{\mathbf{u}}_i^{(t)}$ groundings of the $i$-th formula.

**Proposition 1.** *Using the importance weights shown in Eq. (7) in a normalized estimator (see Eq. (2)) yields an asymptotically unbiased estimate of the query marginals, i.e., as the number of samples, $T \to \infty$, the estimated marginal probabilities almost surely converge to the true marginal probabilities.*

We skip the proof for lack of space, but the idea is that for each unique sample of the outer sampler, each of the importance weight estimates computed using a subset of formula groundings converge towards the true importance weights (if all groundings of formulas were used). Specifically, the weights computed by the "inner" sampler by considering partial groundings of formulas add up to the true weight as $T \to \infty$ and therefore each importance weight is asymptotically unbiased. Eq. (2) is thus a ratio of asymptotically unbiased quantities and the above proposition follows.

We now show how we can leverage MLN structure to improve the weight estimate in Eq. (7). Specifically, we Rao-Blackwellize the "inner" sampler as follows. We partition the variables in each formula into two sets, $\mathbf{V_1}$ and $\mathbf{V_2}$, such that we sample a grounding for the variables in $\mathbf{V_1}$ and for each sample, we tractably compute the exact number of satisfied groundings for all possible groundings to $\mathbf{V_2}$. We illustrate this with the following example.

**Example 1.** *Consider a formula $\neg R(x, y) \vee S(y, z)$ where each variable has domain-size equal to $d$. The data-complexity of computing the satisfied groundings in this formula is clearly $d^3$. However, for any specific value of $y$, say $y = A$, the satisfied groundings in this formula can be computed in closed form as, $n_1 d + n_2 d - n_1 n_2$, where $n_1$ is the number of false groundings of $R(x, A)$ and $n_2$ is the number of true groundings in $S(A, z)$. Computing this for all possible values of $y$ has a complexity of $O(d^2)$.*

Generalizing the above example, for any formula $f$ with variables $\mathbf{V}$, we say that $V' \in \mathbf{V}$ is *shared*, if it occurs more than once in that formula. For instance, in the above example $y$ is a shared variable. Sarkhel et. al [14] showed that for a formula $f$ where no terms are shared, given an assignment to its ground atoms, it is always possible to compute the number of satisfied groundings of $f$ in closed form. Using this, we have the following proposition.

**Proposition 2.** *Given assignments to all ground atoms of a formula $f$ with no shared terms, the combined complexity of computing the number of satisfied groundings of $f$ is $O(d^K)$, where $d$ is an upper-bound on the domain-size of the non-shared variables in $f$ and $K$ is the maximum number of non-shared variables in an atom of $f$.*

Algorithm 1 illustrates our complete sampler. It assumes $\hat{\mathcal{M}}$ and $\zeta$ are provided as input. First, we construct the symbolic equation Eq. (5) that computes the weight of the proposal. In the outer sampler, we sample the symbols from Eq. (5) using Gibbs sampling. After the chain has mixed, for each sample from the outer sampler, for every formula in $\mathcal{M}$, we construct an inner sampler that uses Rao-Blackwelization to approximate the sample weight. Specifically, for a formula $f$, we sample

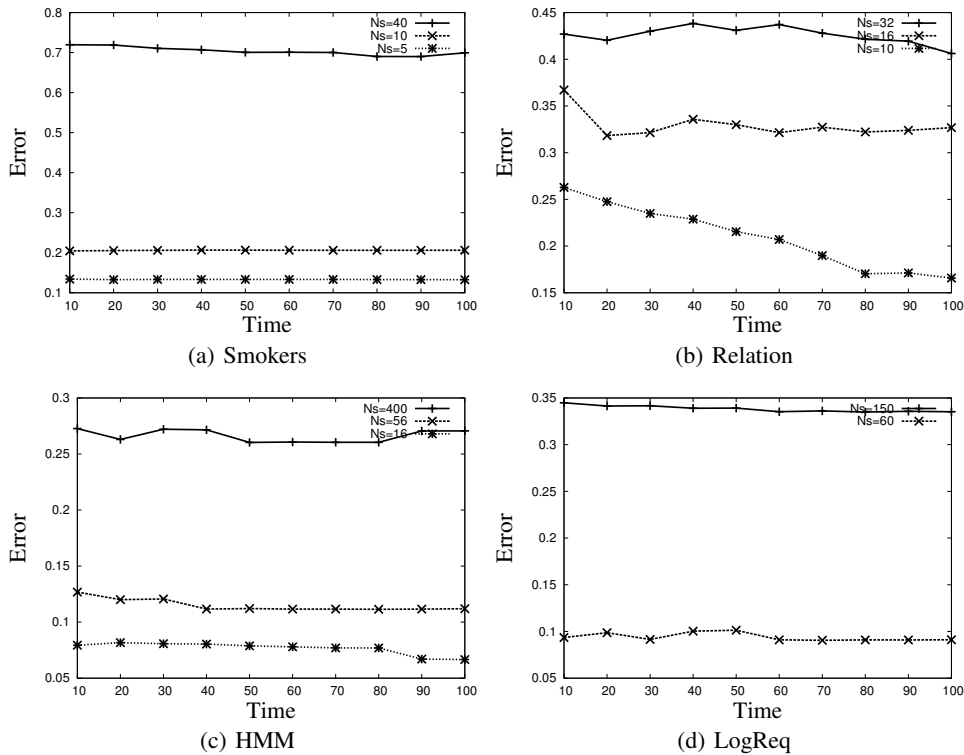

Figure 2: Tradeoff between computational efficiency and accuracy. The y-axis plots the average KL-divergence between the true marginals and the approximated ones for different values of $N_s$. Larger $N_s$ implies weaker proposal, faster sampling. For this experiment, we set $\beta$ (sampling bound) to 0.2. Note that changing $\beta$ did not affect our results very significantly.

an assignment to each non-shared variable to create a partially ground formula, $f'$ and compute the exact number of satisfied groundings in $f'$. Finally, we compute the sample weight as in Eq. (7) and update the normalized estimator in Eq. (2).

## 4 Experiments

We run two sets of experiments. First, to illustrate the trade-off between accuracy and complexity, we experiment with MLNs which can be solved exactly. Our test MLNs include Smokers and HMM (with few states) from the Alchemy website [10] and two additional MLNs, Relation ($R(x, y) \Rightarrow S(y, z)$), LogReq (randomly generated formulas with singletons). Next, to illustrate scalability, we use two Alchemy benchmarks that are far larger, namely Hypertext classification with 1 million ground formulas and Entity Resolution (ER) with 8 million ground formulas. For all MLNs, we randomly set 25% groundings as true and 25% as false. For clustering, we used the scheme in [19] with KMeans++ as the clustering method. For Gibbs sampling, we set the thinning parameter to 5 and use a burn-in of 50 samples. We ran all experiments on a quad-core, 6GB RAM, Ubuntu laptop.

Fig. 2 shows our results on the first set of experiments, where the y-axis plots the average KL-divergence between the true marginals for the query atoms and the marginals generated by our algorithm. The values are shown for varying values of $N_s = \frac{|G_{\mathcal{M}}|}{|G_{\tilde{\mathcal{M}}}|}$, i.e. the ratio between the ground MLN-size and the proposal MLN-size. Intuitively, $N_s$ indicates the amount by which $\mathcal{M}$ has been compressed to form the proposal. As illustrated in Fig. 2, as $N_s$ increases, the accuracy becomes lower in all cases because the proposal is a weaker approximation of the true distribution. However, at the same time, the complexity decreases allowing us to trade-off accuracy with efficiency. Further, the MLN-structure also determines the proposal accuracy. For example, LogReg that contains singletons yields an accurate estimate even for high values of $N_s$, while, for Relation, a smaller $N_s$ yields such

| $(N_s, \beta)$ | C-Time(secs) | I-SRate | $(N_s, \beta)$ | C-Time(secs) | I-SRate |
|---|---|---|---|---|---|
| $(2^{10}, 0.1)$ | 3 | 1200 | $(10K, 0.1)$ | 25 | 125 |
| $(2^{10}, 0.25)$ | 3 | 250 | $(10K, 0.25)$ | 65 | 45 |
| $(2^{10}, 0.5)$ | 3 | 150 | $(10K, 0.5)$ | 65 | 15 |
| $(2^5, 0.1)$ | 8 | 650 | $(1K, 0.1)$ | 65 | 125 |
| $(2^5, 0.25)$ | 8 | 180 | $(1K, 0.25)$ | 65 | 45 |
| $(2^5, 0.5)$ | 8 | 100 | $(1K, 0.5)$ | 65 | 15 |
| $(2^3, 0.1)$ | 15 | 600 | $(2^5, 0.1)$ | 150 | 15 |
| $(2^3, 0.25)$ | 15 | 150 | $(2^5, 0.25)$ | 150 | 8 |
| $(2^3, 0.5)$ | 15 | 90 | $(2^5, 0.5)$ | 150 | 4 |

(a) Hypertext (1M groundings)  (b) ER (8M groundings)

Figure 3: Scalability experiments. C-Time indicates the time in seconds to generate the proposal. I-SRATE is the sampling rate measured as samples/minute.

accuracy. This is because, singletons have symmetries [4, 7] which are exploited by the clustering scheme when building the proposal.

Fig. 3 shows the results on the second set of experiments where we measure the computational-time required by our algorithm during all its operational steps namely proposal creation, sampling and weight estimation. Note that, for both the MLNs used here, we tried to compare the results with Alchemy, but we were unable to get any results due to the grounding problem. As Fig. 3 shows, we could scale to these large domains because, the complexity of sampling the proposal is feasible even when generating the ground MLN is infeasible. Specifically, we show the time taken to generate the proposal distribution (C-Time) and the the number of weighted samples generated per minute during inference (I-SRate). As expected, decreasing $N_s$, or increasing $\beta$ (sampling bound) lowers I-SRate because the complexity of sampling increases. At the same time, we also expect the quality of the samples to be better. Importantly, these results show that by addressing the evidence/grounding problems, we can process large, arbitrarily structured MLNs/evidence without running out of memory in a reasonable amount of time.

## 5 Conclusion

Inference algorithms in Markov logic encounter two interrelated problems hindering scalability – the grounding and evidence problems. Here, we proposed an approach based on importance sampling that avoids these problems in every step of its operation. Further, we showed that our approach yields asymptotically unbiased estimates. Our evaluation showed that our approach can systematically trade-off complexity with accuracy and can therefore scale-up to large domains.

Future work includes, clustering strategies using better similarity measures such as graph-based similarity, applying our technique to MCMC algorithms, etc.

**Acknowledgments**

This work was supported in part by the AFRL under contract number FA8750-14-C-0021, by the ARO MURI grant W911NF-08-1-0242, and by the DARPA Probabilistic Programming for Advanced Machine Learning Program under AFRL prime contract number FA8750-14-C-0005. Any opinions, findings, conclusions, or recommendations expressed in this paper are those of the authors and do not necessarily reflect the views or official policies, either expressed or implied, of DARPA, AFRL, ARO or the US government.

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
