[Reviews · NeurIPS 2014]

Submitted by Assigned_Reviewer_10

This paper combines evidence-based clustering with an importance sampling approach. It further optimizes the weight function calculation of importance sampling, by using a second sampler to estimate the number of true groundings.

I recommend to accept this paper because of the importance of the evidence problem, and the fact that the proposed solution is very general and scalable.

The paper is well-written and clear, except for Figure 1 (b) and Equations 5,6, and 7, which are poorly explained.

A disadvantage of the paper is that its clustering algorithm is reused from a recent publication:
"Evidence-based Clustering for Scalable Inference in Markov Logic" at ECML 2014.
Thus, the clustering algorithm is not a contribution of this paper.
Another disadvantage is the complexity of the setup.

The introduction says that the combination of the base importance sampler, and the sampler for the number of groundings yield unbiased estimates. This claim is not supported by the main text. The first sampler is obviously unbiased when using the exact number of groundings. Proposition 2 says that the second sampler is unbiased. I don't think this implies that the combination of the two is unbiased. Even if the second sampler is unbiased, it has a variance because it uses a finite fixed sample size. This variance can induce a bias in the first sampler?

The experimental setup is impressive, because of the scale of the MLN models considered. The results are slightly disappointing though: the errors do not go down significantly. Is this to be expected?

Beta is used in line 300 and 400/Fig3 for different variables? Is the second beta explained?

Summary: This paper attempts to solve two important problems in lifted inference: grounding and asymmetric evidence. The proposed solution is general and novel, and will have an impact on the lifted inference community. A weakness is the complexity of the solution, involving multiple samplers, and a strong similarity to a recently published paper on evidence-based clustering.

Submitted by Assigned_Reviewer_21

The paper proposes an importance sampling algorithm for Markov logic networks that scales up even when few symmetries are present. The idea is to cluster the constants and use the cluster averages in the proposal distribution. The paper describes in detail the choice of clustering algorithm and distance measure, how to sample from the proposal, and how to approximately calculate the important weights.

The proposed method is the first to deal with scalability with few symmetries besides [19]. It is more general than [19] because [19] can only handle binary evidence. Similar to [19], it has parameters (Ns, beta) to trade off between efficiency and accuracy, so theoretically it handles very large MLNs with arbitrary evidence.

The experiments of the paper can be improved. First, although the method is more general than [19], since the datasets used in the experiments have only binary evidence, it would be good to have a comparison of accuracy and complexity with [19]. Second, in the second experiment (large MLNs), it would be more convincing to use the actual truth values of evidence in the datasets of Alchemy, instead of using randomly sampling evidence (25% true and 25% false). In this way, the accuracy can also be evaluated against the ground truth, so the trade off between accuracy and efficiency can be illustrated on large MLNs. Third, the influence of beta on accuracy is never mentioned, which should be discussed in some way.
Summary: The paper proposes a scalable lifted importance sampling for MLN by trading off complexity with accuracy. The problem solved by this paper is important and practical (scaling up MLN inference). The approach is very novel and well presented. The experiments show that the method is effective on very large MLNs but not very convincing. It is the largest weakness of the paper.

Submitted by Assigned_Reviewer_42

This paper proposes a scalable importance sampling to avoid grounding the full MLN and focuses on approximations of marginal probabilities of query atoms given evidence.
It mentions complex structure and evidence as the two main reasons to break symmetries in MLNs that causes problems for scalability in lifted inference methods.

In order to make the overall importance sampling process tractable each steps is made tractable in this approach.
The method designs a proposal distribution from a compressed MLN representation via clustering of related groundings, by defining a partitioning.
Then it uses Gibbs sampling and show that the sampling space is proportional to the number of chosen clusters. To estimate the importance weight it samples a bounded subset of ground formulas in MLNs.

The paper shows that the proposed approach yields asymptotically unbiased estimate, and in the experimental evaluation the trade off between complexity and accuracy, has been demonstrated.

In the experimental evaluation two sets of experiments are reported; one to compare marginal probabilities based on the proposed sampling approach with the exact values (using Alchemy), and the other to demonstrate scalability with larger datasets.

While the paper provides a brief summary of related works that focus on scaling MLNs or probabilistic relational models in general, it does not provide a performance comparison with any other methods that aim to scale MLNs in the evaluation section.
Summary: The paper proposes an interesting method to scale-up importance samplings for MLNs but it does not provide a performance comparison with any other methods that aim to scale MLNs in the evaluation section.
I think the authors should clearly lay out the boundaries of current approaches in scaling up MLNs (in terms of accuracy, size of data, and speed) and show which of these three they have improved and by how much.
Author Feedback
Author rebuttal: We thank all the reviewers for their insightful comments

Assigned_Reviewer_10
Yes, the design of the proposal partially uses the evidence-based clustering idea presented in the ECML paper. Specifically as in the ECML paper, "similar" domains are clustered together. However, the main problem with the ECML paper is that it is inherently heuristic i.e. it improves scalability by reducing the domain-size of the MLN but at the same time, this domain-reduction alters the distribution represented by the MLN. In some cases (depending on the clustering/distance-function quality), it is possible that the new distribution is quite far off from the MLN's actual distribution which is undesirable. In contrast, we believe our work here is a significant improvement on the ECML work because it is much more general, systematic and has guarantees; we never change the MLN distribution but still scale-up inference to very large domain-sizes.

We will try to simplify the equations 5,6,7 and the example.

Regarding the complexity of the setup, we wanted to make the system more general and applicable for all MLNs/evidence while maintaining scalability; this is quite challenging. Therefore, we had to fix scalability issues in every step which makes the technical details/setup seem complex. However, the basic idea in our approach is not that hard to grasp: 1) Utilize off-the-shelf clustering algorithms (along with a distance measure) for constructing the proposal. 2) Sample the proposal using a tractable sampling algorithm (we have used Gibbs sampling but variational methods such as SampleBP can be also be applied) and 3) Approximate the weight of each sample using a tractable algorithm (while maintaining some desirable properties).

Note that we are not claiming “unbiased estimates.” We are claiming “asymptotically unbiased estimates.” (Although, we wrote the paper for computing the marginal probabilities and not the partition function, the following applies in a straight-forward manner to the partition function. Conventional/Lifted importance sampling yields an unbiased estimate of the partition function. However, our method will only yield an asymptotically unbiased estimate of the partition function).

In particular, as mentioned in the theorems and propositions, the combination of the two samplers will not be unbiased but only asymptotically unbiased which is a weaker guarantee. That is, only when the sample size is very large (infinity), the expected value of the sample estimates approaches the true values. This means that as you mention, even though the "inner" sampler biases the result of the full sampler for each sample, overall as the number of samples go to infinity and the sample-weights add up, the estimates are asymptotically unbiased. Moreover, the variance of both samplers will be zero for infinite sample size (variance is inversely proportional to the number of samples).

Experiments:
Yes, the second beta is not specified correctly, it is the percentage of total groundings. We will correct this.

The errors largely depended upon the proposal which in turn depends on the structure of the MLN as well as the performance of the clustering/distance-function given that specific evidence. We are trying to study the "types" of MLNs/evidence for which we can generate strong proposals maybe using more complex distance functions. This is part of our future work.

Assigned_Reviewer_21

Regarding [19], there was no easily available implementation on the author's website(http://dtai.cs.kuleuven.be/ml/systems/wfomc did not list the evidence approximation). Therefore, we could not compare with this empirically.

Regarding experimenting jointly with accuracy and scalability, even if we use the evidence data as specified by Alchemy, we are not sure how to compute the true marginal probabilities (of non-evidence atoms) because this is not computable exactly by any existing system as far as we know.

In our experiments, changing beta did not have as large an influence on the accuracy. Therefore, we did not include the graphs for lack of space. We will include a note on this.

Assigned_Reviewer_42

The problem of inference in MLNs with arbitrary structure/evidence is extremely challenging and therefore there are not too many publically available systems capable of this (there are various papers that we cited but no publicly available implementations that we can compare with). We tried to compare our system with Alchemy and Alchemy 2.0 which are two well-known state-of-the-art MLN inference systems. Particularly, Alchemy 2.0 has a number of "lifted" algorithms that are specifically designed to scale-up inference to large domain-sizes. However, they work only when given the right MLN and evidence structure. Therefore, as we mention in the paper, we were not able to generate any meaningful comparison because the inference algorithms in Alchemy and Alchemy 2.0 were not able to scale-up when given arbitrary MLNs/evidence structure. We will try to make this observation much more explicit in our paper.

There is some recent work on scaling-up inference in the presence of evidence by approximating evidence [19] but its implementation was not publicly available (please see response to Assigned_Reviewer_21).